# The Role of Copper in the Regulation of Ferroportin Expression in Macrophages

**DOI:** 10.3390/cells10092259

**Published:** 2021-08-31

**Authors:** Aneta Jończy, Rafał Mazgaj, Ewa Smuda, Beata Żelazowska, Zuzanna Kopeć, Rafał Radosław Starzyński, Paweł Lipiński

**Affiliations:** Department of Molecular Biology, Institute of Genetics and Animal Biotechnology, Polish Academy of Sciences, 05-552 Jastrzębiec, Poland; r.mazgaj@igbzpan.pl (R.M.); e.smuda@igbzpan.pl (E.S.); b.zelazowska@igbzpan.pl (B.Ż.); z.kopec@igbzpan.pl (Z.K.); r.starzynski@igbzpan.pl (R.R.S.)

**Keywords:** ferroportin, copper, macrophages, Nrf2, LPS

## Abstract

The critical function of ferroportin (Fpn) in maintaining iron homeostasis requires complex and multilevel control of its expression. Besides iron-dependent cellular and systemic control of Fpn expression, other metals also seem to be involved in regulating the *Fpn* gene. Here, we found that copper loading significantly enhanced *Fpn* transcription in an Nrf2-dependent manner in primary bone-marrow-derived macrophages (BMDMs). However, prolonged copper loading resulted in decreased Fpn protein abundance. Moreover, CuCl_2_ treatment induced Fpn expression in RAW 264.7 macrophages at both the mRNA and protein level. These data suggest that cell-type-specific regulations have an impact on Fpn protein stability after copper loading. Transcriptional suppression of Fpn after lipopolysaccharide (LPS) treatment contributes to increased iron storage inside macrophages and may result in anemia of inflammation. Here, we observed that in both primary BMDMs and RAW 264.7 macrophages, LPS treatment significantly decreased Fpn mRNA levels, but concomitant CuCl_2_ stimulation counteracted the transcriptional suppression of Fpn and restored its expression to the control level. Overall, we show that copper loading significantly enhances *Fpn* transcription in macrophages, while Fpn protein abundance in response to CuCl_2_ treatment, depending on macrophage type and factors specific to the macrophage population, can influence Fpn regulation in response to copper loading.

## 1. Introduction

Ferroportin (Fpn) is the only known mammalian iron exporter, and it plays a vital role in the process of iron absorption from duodenal enterocytes and the reutilization of this microelement from senescent erythrocytes by reticuloendothelial macrophages. Due to its unique function in iron metabolism, Fpn expression remains under tight and complex control at the transcriptional, post-transcriptional, and posttranslational level [1]. Moreover, local and cell-type-specific Fpn regulation seems to be critical for maintaining iron homeostasis [2].

Fpn post-transcriptional regulation involves the interaction of iron regulator proteins (IRP1/2) with iron responsive element (IRE) sequences within the 5′ UTR of Fpn mRNA [3]. At the protein level, Fpn is strongly downregulated by hepcidin (a peptide produced mainly by the liver in response to increased iron content), which triggers its internalization and degradation [4]. Transcriptional Fpn regulation involves the heme-dependent binding of transcriptional repressors (Bach1) or activators (Nrf2) to antioxidant response element (ARE) sequences localized within the *Fpn* promoter, highlighting the role of heme in the process [5,6]. It has been shown that iron itself can also increase *Fpn* transcription in macrophages and lung epithelial cells [7,8,9]. Interestingly, *Fpn* expression in macrophages was also increased upon loading with other metals, such as cadmium, cobalt, copper, manganese, and zinc. Zn and Cd are able to induce *Fpn* transcription by eliciting metal transcription factor-1 (MTF-1) translocation into the nucleus [9]. In macrophages, copper can stimulate Fpn expression at both the mRNA and protein level, as has been shown in the J774 macrophage cell line [10]. Although biological copper-iron interactions involved in processes of absorption, storage, and redistribution of these metals were described years ago [11], their detailed molecular mechanisms are still not fully elucidated. Therefore, the first aim of this study was to determine the possible molecular mechanisms of the regulation of Fpn expression by copper in macrophages, which plays a crucial role in systemic iron turnover.

Macrophages are pivotal components of the immune system, serving as key modulators and effectors of immune response, and are highly present in the chronic and acute inflammatory milieu [12]. One of the specific features of pro-inflammatory macrophages is iron retention within the cells as a result of the downregulation of Fpn expression and increased intracellular levels of ferritin, an iron storage protein [13,14]. This mechanism leads to reduced iron concentration in the plasma, and thus limits iron availability for extracellular pathogens, but at the same time decreases the iron supply for erythropoiesis, which contributes to the appearance of so-called anemia of infection (AI) or anemia of chronic diseases (ACD), the most prevalent anemias in hospitalized patients and the second most frequent type after iron-deficiency anemia [15,16]. Lipopolysaccharide (LPS), a component of the cell wall of Gram-negative bacteria, triggers the release of inflammatory cytokines in macrophages [17]. Previous studies reported that LPS treatment significantly downregulated *Fpn* expression in monocytic cells [18], but the mechanism is only partially understood. Interestingly, Nrf2 activation (transcription factor, nuclear factor, erythroid 2 like 2) during inflammation can counteract lipopolysaccharide-induced suppression of Fpn mRNA and enhance iron efflux in macrophages [19], suggesting that the regulation of *Fpn* expression by Nrf2 may contribute to anti-inflammatory processes.

It is well known that serum copper levels increase under a variety of inflammatory conditions [20,21], and stimulating macrophages with LPS significantly influences the expression of copper-related genes, such as cellular copper importer (Ctr1) and the P1B-type ATPase responsible for delivering copper into cellular compartments (Atp7a) [22,23]. This study aims to illuminate the role of copper in the regulation of Fpn expression in macrophages and test the hypothesis that assumes the stimulation of Fpn expression by Cu ions at least partially counterbalances the mechanism leading to decreased Fpn levels and, consequently, iron retention within macrophages under LPS activation.

## 2. Materials and Methods

### 2.1. Cell Culture

#### 2.1.1. Bone-Marrow-Derived Macrophages (BMDMs)

BMDMs were isolated from tibias, femurs, and humeri of 3-to-5-month-old male wild-type (C57Bl/6J) or Nrf2 knockout (B6.129CX1-Nfe2l2tm1Ywk/J) mice derived from Jackson Laboratory (Bar Harbor, ME, USA). Cells were seeded in 10 cm diameter Petri dishes for protein extraction, 6-well plates for RNA isolation, and rounded coverslips for confocal imaging. BMDMs were cultured in RPMI 1640 medium with l-glutamine and 25 mM HEPES buffer (Gibco, Thermo Fisher Scientific, Waltham, MA, USA) supplemented with 10% heat-inactivated FBS (EURx, Gdańsk, Poland), 10% LCCM (L929-cell conditioned medium as a source of macrophage colony-stimulating factor), and 1% penicillin/streptomycin (Sigma-Aldrich, Burlington, MA, USA) at 37 °C, in 5% CO_2_ and 21% O_2_ atmosphere.

#### 2.1.2. Peritoneal Macrophages

Mouse primary peritoneal macrophages were isolated as previously described [24]. Briefly, 3-to-5-month-old male mice were euthanized with CO_2_, followed by surgical exposure of the inner skin lining the peritoneal cavity. Ice-cold PBS (5 mL) supplemented with 3% FCS was injected into the peritoneal cavity using a 27 G needle, then gentle massage of the peritoneum was performed to dislodge attached cells into the PBS solution. The fluid was carefully collected to avoid clogging by fatty tissue or other organs. Cell suspension was centrifuged at 800× *g* for 8 min and cell pellets were resuspended in the culture medium used for BMDM culture. Cells were seeded in 24-well cell culture plates for In-Cell Western blot analysis or 6-well plates for RNA isolation.

#### 2.1.3. RAW 264.7 Cell Line

RAW 264.7 murine macrophages were obtained from the American Type Culture Collection (Rockville, MD, USA). Cells were cultured in DMEM (Gibco, Thermo Fisher Scientific, Waltham, MA, USA) containing 5% (*v*/*v*) FBS and 50 μg/mL gentamicin (Sigma-Aldrich, Burlington, MA, USA) in 175 cm^2^ plastic culture flasks for protein and 6-well plates for RNA isolation.

### 2.2. Cytotoxicity Assay and ROS Production Measurements

The CellTox™ Green Cytotoxicity Assay (Promega, Madison, WI, USA) was used to assess cytotoxic effects of dose- and time-dependent CuCl_2_ stimulation of BMDMs via real-time measurement of changes in membrane integrity occurring as a result of cell death. BMDMs were seeded in 96-well plates, incubated with CuCl_2_ (25–500 μM) for 8–24 h, and mixed with CellTox™ Green Dye according to the manufacturer’s instructions. Fluorescence was measured using the BMG Fluostar microplate reader (Ex: 485 nm, Em: 520 nm).

Reactive oxygen species (ROS) production in BMDMs treated with CuCl_2_ was assessed using a chemically reduced form of fluorescein (H_2_DCFDA) (Thermo Fisher Scientific, Waltham, MA, USA). Upon cleavage of the acetate groups by intracellular esterases and oxidation, the non-fluorescent H_2_DCFDA is converted to fluorescent 2′,7′-dichlorofluorescein (DCF). After CuCl_2_ treatment, BMDMs were incubated with 5 μM H_2_DCFDA probe for 30 min, followed by staining of nuclei with Hoechst 33342 (Thermo Fisher Scientific, Waltham, MA, USA). Coverslips were attached to the microscope slides and imaged on confocal microscope (A1R Nikon) using excitation sources and filters appropriate for fluorescein.

### 2.3. Real-Time Quantitative PCR (RT-qPCR)

Total RNA from BMDMs, peritoneal macrophages, and RAW 264.7 cells was isolated using the SV Total RNA Isolation System (Promega, Madison, WI, USA) according to the manufacturer’s protocol. Then, 800 ng (BMDMs, RAW 264.7 cells) or 200 ng (peritoneal macrophages) of total DNAse-treated RNA was reverse transcribed using a Transcriptor First Strand cDNA Synthesis Kit (Roche Diagnostics, Basel, Switzerland). The amplified products were detected using SYBR Green I (Roche Diagnostics, Basel, Switzerland). Real-time quantitative PCR analysis was performed in a Light Cycler U96 (Roche Diagnostics, Basel, Switzerland) using gene-specific primer pairs (Appendix A). Transcript levels were normalized relative to ribosomal protein L4 (Rpl4), ribosomal protein L19 (Rpl19), or hypoxanthine-guanine phosphoribosyltransferase (Hprt) expression levels. Control reference genes were selected using NomFinder software (NomFinder v0.953, Aarhus, Denmark, https://moma.dk/normfinder-software, accessed on 1 June 2014). Analysis of RT-qPCR results was based on −ΔCt (Ct of reference gene − Ct of test gene).

### 2.4. Protein Extract Isolation and Western Blotting

Cultured BMDMs and RAW 264.7 cells were washed with cold phosphate-buffered saline (PBS), scraped in 2 mM EDTA/PBS buffer, and centrifuged (500× *g*, 5 min, 4 °C). To obtain membrane and cytosolic fractions, cell pellets were suspended in lysis buffer (10 mM Tris HCl, pH 7; 1 mM MgCl_2_) supplemented with a cocktail of protease inhibitors (Sigma-Aldrich, Burlington, MA, USA) and phenylmethanesulfonylfluoride (PMSF; Sigma-Aldrich, Burlington, MA, USA), left on ice for 30 min and homogenized by passing through a 25⅝ G needle. The homogenate was centrifuged (8000× *g*, 5 min, 4 °C) and supernatant was ultracentrifuged (80,000× *g*, 45 min, 4 °C). The supernatants obtained after ultracentrifugation and enriched in cytosolic protein fraction were collected and stored at −80 °C. Final pellets (membrane extract) were resuspended in TNE buffer (10 mM Tris HCl, pH 7; 10 mM EDTA; 0.1 M NaCl) and stored at −80 °C until use.

To obtain total cellular extracts, cell pellets were suspended in RIPA buffer (10 mM Tris pH8, 150 mM NaCl, 1 mM EDTA, 1% Igepal (NP40), 0.1% SDS) supplemented with a protease inhibitor cocktail (Sigma-Aldrich, Burlington, MA, USA) and phenylmethanesulfonylfluoride (PMSF; Sigma-Aldrich, Burlington, MA, USA). The cells were left on ice for 30 min and then centrifuged (6000× *g*, 15 min, 4 °C). Protein concentrations were determined by Bradford assay (Bio-Rad, Hercules, CA, USA) and protein extracts were subjected to SDS-PAGE electrophoresis. Samples were not boiled before loading. Before membrane blocking with 5% skim milk, transfer efficiency was confirmed by Ponceau-S staining. A list of primary and secondary antibodies used is given in Appendix A. After incubation with primary antibody (overnight, 4 °C), blots were washed (TTBS buffer) and incubated with horseradish peroxidase-conjugated secondary antibodies at room temperature for 1 h, followed by visualization with the enhanced luminescence kit (Thermo Fisher Scientific, Waltham, MA, USA).

### 2.5. In-Cell Western Blot

Primary peritoneal macrophages cultured in 24-well plates were fixed with ice-cold methanol at −20 °C for 15 min, washed with PBS, and permeabilized with Triton X-100 (0.1%/PBS) for 10 min. Cells were blocked with Odyssey blocking buffer (LI-COR Bioscience, Lincoln, NE, USA) for 90 min at room temperature. Cells were incubated overnight at 4 °C with rabbit anti-mouse Fpn antibody (1/500; kind gift from F. Canonne-Hergaux, INSERM, Toulouse, France) and mouse anti-mouse β-actin (1:500, Thermo Fisher Scientific, Waltham, MA, USA) in Odyssey blocking buffer. After being washed with 0.5% Tween-20/PBS, cells were incubated for 1 h at room temperature with donkey anti-rabbit LI-COR IRDye 680RD labeled secondary antibody (red, 1/500; LI-COR Bioscience, Lincoln, NE, USA) and donkey anti-mouse LI-COR IRDye 800CW labeled secondary antibody (green, 1/500; LI-COR Bioscience, Lincoln, NE, USA). Cells were washed 5 times with 0.5% Tween-20/PBS and scanned by the Odyssey^®^ Infrared Imaging System.

### 2.6. Isolation of Lipid Raft Fractions

Lipid rafts from RAW 264.7 cells were isolated as described previously [25,26]. Briefly, cells were washed with ice-cold PBS, scraped in 2 mM EDTA/PBS, centrifuged (500× *g*, 5 min, 4 °C), and incubated on ice for 30 min in 600 μL lysis buffer (150 mM NaCl; 25 mM MES; 5 mM EDTA, pH 6.5; 1% Triton X-100) supplemented with a cocktail of protease inhibitors (Sigma-Aldrich, Burlington, MA, USA) and phenylmethanesulfonylfluoride (PMSF; Sigma-Aldrich, Burlington, MA, USA). Samples were homogenized by passing through a 25⅝ G needle. The homogenate was centrifuged (8000× *g*, 5 min, 4 °C) and supernatant was adjusted to a final concentration of 40% (*w*/*v*) iodixanol (OptiPrep^®^, Sigma-Aldrich, Burlington, MA, USA), and the mixture was then layered under a 20–40% discontinuous iodixanol gradient and centrifuged at 260,000× *g* for 16 h at 4 °C using an SW 41 Ti Rotor (Beckman Coulter, Brea, CA, USA). Then, 11 fractions of 1 mL were collected from the top to the bottom of the gradient tube, and Western blot analysis of the fractions was performed. Lipid raft fractions were defined by the presence of a raft marker, flotilin 2 [27], while TfR1 was considered as a marker of non-lipid raft domains [28].

### 2.7. Chemiluminescent RNA Electrophoretic Mobility Gel Shift Assay (EMSA)

RNA-EMSA reaction was performed using a LightShift^®^ Chemiluminescent RNA EMSA Kit (Thermo Fisher Scientific, Waltham, MA, USA) according to the manufacturer’s instructions. Briefly, cytosolic extracts from control and CuCl_2_- or LPS-treated BDMDs were obtained using NE-PER™ Nuclear and Cytoplasmic Extraction Reagents (Thermo Fisher Scientific, Waltham, MA, USA). Then, 20 μg of cytosolic BMDM extract was mixed with REMSA Binding Buffer, glycerol, tRNA, and Biotin-IRE RNA probe and incubated for 30 min at room temperature. Samples were loaded on 5% native polyacrylamide gel and subjected to protein electrophoresis in non-reducing and non-denaturing conditions in 0.5× TBE buffer (100 V, 1 h, 4 °C). Electrophoresed samples were transferred to a positively charged nylon membrane (400 mA, 30 min, 4 °C) (Thermo Fisher Scientific, Waltham, MA, USA). After transfer, crosslinking of RNA to membrane was performed using a UV lamp with 254 nm bulbs for 4 min. Membrane was blocked with the manufacturer’s blocking buffer and incubated with stabilized streptavidin-horseradish peroxidase conjugate for 15 min and washed extensively, and signal was developed using the chemiluminescence detection method. As a control, biotin-IRE control RNA and cytosolic liver extract +200-fold molar excess of unlabeled IRE probe were used.

### 2.8. Statistical Analysis

Data are presented as mean values ± standard deviation (SD). Statistical analysis was performed using Student’s *t*-test (two groups considered) or one-way ANOVA (multiple groups) followed by Dunnett’s post hoc multiple comparison test. Two-way ANOVA was used to assess statistical significance of the cytotoxic effect of CuCl_2_ treatment on BMDMs, followed by Tukey post hoc test. Statistical analysis was performed using GraphPad Prism 8.4.2 software (GraphPad, San Diego, CA, USA). * *p* ≤ 0.05, ** *p* ≤ 0.01, *** *p* ≤ 0.001, and **** *p* ≤ 0.0001 were considered significant.

## 3. Results

### 3.1. Exposure of Bone-Marrow-Derived Macrophages to CuCl_2_ Results in Intracellular Copper Loading without Significant ROS Production

The toxicity of copper is usually associated with its redox activity. In order to determine a nontoxic dose of CuCl_2_ and the optimal timing of exposure of bone-marrow-derived macrophages (BMDMs) to Cu ions, cell viability and cytotoxicity assays were performed after the incubation of BMDMs with increasing concentrations of CuCl_2_. We observed that only a high CuCl_2_ concentration (500 μM) elicited strong cytotoxicity compared to lower doses, regardless of the duration of exposure (Figure 1a). Overnight (24 h) incubation of BMDMs with 150 μM CuCl_2_ significantly increased the cellular copper content (as measured by ICP-MS), indicating effective copper transport into the cells (Figure 1b). The intracellular reactive oxygen species (ROS) level was measured by the intensity of fluorescence of H_2_DCFDA-treated cells. The data demonstrate that despite increased intracellular Cu loading, BMDMs treated with 150 μM CuCl_2_ did not show increased ROS production compared to cells exposed to cytotoxic 500 μM CuCl_2_ (Figure 1c).

### 3.2. Copper Loading Upregulates Fpn mRNA Abundance in BMDMs in an Nrf2-Dependent Manner

Previous studies have shown that copper increases Fpn mRNA levels in the J774 macrophage cell line, but the detailed mechanism of this regulation has not been defined [10]. Here, we report that in primary BMDMs, Fpn mRNA expression increased significantly after exposure to CuCl_2_ in a dose- and time-dependent manner (Figure 2a,b). To determine whether copper loading affects the transcription rate and Fpn mRNA stability, BMDMs were pretreated with 150 μM CuCl_2_ for 16 h and exposed for 2 and 8 h to an RNA synthesis inhibitor, α-amanitin (2 μg/mL). As shown in Figure 2c, α-amanitin blocked copper-induced Fpn mRNA expression but significantly increased its stability, as demonstrated by higher mRNA levels after 8 h incubation with the inhibitor.

In separate studies, heme was shown to be a strong positive transcriptional regulator of Fpn mRNA expression, which allows the binding of Nrf2 to the ARE sequence within the Fpn promoter [5].

The incubation of BMDMs with 150 μM CuCl_2_ for 16 h led to an increased expression of Fpn and Nqo1, the previously described Nrf2 target gene (Figure 2d,e) [29]. Moreover, as shown in Figure 2f, the basal level of Fpn expression strictly depends on Nrf2. Interestingly, the CuCl_2_ transcriptional effect was not observed in BMDMs derived from Nrf2 knockout mice (Figure 2g–i), which clearly indicates the role of Nrf2 in copper-dependent transcriptional Fpn upregulation.

### 3.3. Copper Loading Counterbalances LPS-Induced Suppression of Fpn mRNA Expression in Macrophages

It has been shown previously that LPS treatment decreases Fpn expression, and pharmacological activators of Nrf2 can counteract the LPS-induced suppression of *Fpn* mRNA [19,30]. Our results, as well as studies on BMDMs described by Achard and co-workers [22], indicate that LPS treatment significantly increases the expression of genes responsible for copper uptake (Ctr1) and intracellular trafficking (Atp7a) in BMDMs (Figure 3a,b). Interestingly, while LPS stimulation leads to a profound decrease in Fpn expression, concomitant CuCl_2_ treatment restores this declined expression to the level observed in control BMDMs (Figure 3c). Moreover, the same effect was observed in RAW macrophages (Figure 3d). These data indicate that the copper-dependent transcriptional upregulation of Fpn mRNA expression is effective enough to counteract LPS-mediated downregulation.

### 3.4. Possible Mechanisms of Copper-Dependent Decrease in Fpn Protein Level

In order to confirm that the upregulation of Fpn mRNA expression is reflected in higher protein levels in BMDMs, Western blot analysis using subcellular membrane fraction was performed. Surprisingly, the exposure of BMDMs to 150 μM CuCl_2_ for 16 h did not significantly affect Fpn protein levels (Figure 4a, upper panel). Moreover, longer incubation of BMDMs with CuCl_2_ (24–48 h) led to a marked decrease in Fpn abundance on macrophage membranes (Figure 4a, bottom panel). To assess whether the reduced level of this iron exporter could affect cellular iron homeostasis, the level of ferritin, an iron storage protein, was evaluated. Cytosolic ferritin is composed of 24 subunits of two types, light (L-Ft) and heavy (H-Ft), depending on their relative molecular weights. The H chain exhibits ferroxidase activity and is especially important for macrophages [31,32]. As shown in Figure 4b,c, Fpn downregulation did not significantly change the level of either L- or H-Ft subunits. Moreover, CuCl_2_ treatment had no influence on the expression of transferrin receptor-1 (TfR1) mRNA (Figure 4d). The TfR1 mRNA level reflects intracellular iron content, as its abundance is inversely correlated with cellular iron status by means of post-transcriptional regulation through Iron Regulatory Proteins (IRPs) that bind iron-responsive element (IRE) sequences located within the 3′UTR of TfR1 mRNA [33,34].

Hepcidin is the main regulator of iron homeostasis. Its binding to ferroportin leads to the rapid internalization and degradation of this protein [4]. Hepcidin is mainly produced by hepatocytes, but its expression increases in pro-inflammatory macrophages infected with intracellular pathogens [35]. Indeed, the exposure of BMDMs to 100 ng/mL LPS significantly increased hepcidin mRNA levels, whereas treatment with 150 μM CuCl_2_ did not change the expression of the Hamp gene (Figure 4e,f). This allows us to conclude that the hepcidin-ferroportin interaction is not responsible for the decrease in the Fpn protein level after prolonged CuCl_2_ treatment.

The translational regulation of Fpn occurs via the binding of Iron Regulatory Proteins (IRPs) to iron responsive element (IRE) sequences localized within the 5′ UTR of the Fpn transcript [3,36]. In order to determine if CuCl_2_ treatment influences IRE-IRP binding activity, we performed an electrophoretic mobility shift assay (EMSA) in BMDM cytosolic extracts. As shown in Figure 4g, we did not observe significant differences in IRP-IRE binding activity between control and CuCl_2_-treated cells. Moreover, LPS treatment significantly increased IRP-IRE binding activity, which was previously shown in the J774 macrophage cell line [37]. Similarly, the exposure of cells to CuCl_2_ did not influence this activity, which suggests that the mechanism responsible for the decreased Fpn protein level in copper-loaded BMDMs does not involve translational repression through the IRP/IRE system.

Lysosomal proteolytic breakdown of Fpn is thought to be responsible for its biological degradation [4]. To further investigate the mechanism responsible for membrane-bound Fpn protein CuCl_2_-mediated downregulation, we treated BMDMs with 100 μM chloroquine (Chlq), a known inhibitor of lysosome function. Surprisingly, Chlq treatment led to a significant decrease in Fpn abundance in the membrane fraction of BMDMs (Figure 5a). Moreover, Chlq and concomitant CuCl_2_ treatment affected cellular ferritin protein levels, leading to increased abundance of the H-Ft subunit (Figure 5b, upper panel), accompanied by a persistent Fpn decline (Figure 5b, bottom panel). We checked whether CuCl_2_ treatment affects the expression of genes involved in the processes of ubiquitination (Usp14, Ubc), multivesicular bodies (MVBs; specialized subset of endosomes that contain membrane-bound intraluminal vesicles) (Lip5, Chmp5), and proteasome formation (Psma1, Psmc4), and did not observe statistical differences (Figure 5c–e).

### 3.5. CuCl_2_ Treatment Affects Fpn Expression in Different Subset of Macrophages

We further investigated whether, similar to BMDMs, exposure of the RAW 264.7 monocyte/macrophage-like cell line and primary peritoneal macrophages to CuCl_2_ would affect Fpn expression. As shown in Figure 6a, CuCl_2_ treatment slightly increased Fpn mRNA expression (*p* = 0.05) in peritoneal macrophages, but in contrast to BMDMs, the Fpn protein level was not changed after CuCl_2_ exposure, as evaluated by In-Cell Western blot quantification (Figure 6b). This indicates that factors specific to the macrophage population can influence Fpn regulation in response to copper loading.

Interestingly, treatment with 150 µM CuCl_2_ led to a significant time-dependent increase in Fpn expression in RAW 264.7 macrophages, on both the mRNA and protein level (Figure 7a,b). Interestingly, the upregulation of Fpn mRNA was already significant after 6 h exposure to 150 μM CuCl_2_ (data not shown), which was not the case for BMDMs. Increased Fpn protein abundance was observed in both total cell extract and membrane subcellular fractions (Figure 7b). It has been previously described that in macrophages, ferroportin is mostly detected in detergent-resistant membranes containing lipid raft markers, and iron overload strongly increases the presence of ferroportin in the lightest raft fraction [25]. Here, we investigated whether copper loading leads to similar Fpn protein distribution within membrane fractions as observed after iron treatment. Raft-enriched fractions from RAW macrophages were obtained by Optiprep gradient density ultracentrifugation, followed by Western blot analysis of membrane fractions. As shown in Figure 7c, in RAW macrophages, after iron treatment, Fpn was mostly detected in lighter lipid raft-enriched membrane fractions. Interestingly, exposure to CuCl_2_ led to Fpn localization within lipid raft and non-lipid raft-enriched fractions, which could reflect the distinct regulation of Fpn present on the cellular membrane.

## 4. Discussion

Reticuloendothelial macrophages (located in the liver, spleen, and bone marrow) play a crucial role in systemic iron turnover [38]. These cells express a set of proteins strictly connected to iron metabolism, including membrane-bound ferroportin (Fpn), the only known iron exporter. Due to its critical function in iron metabolism, Fpn expression is tightly regulated at both the cellular and systemic level. Most of the regulatory mechanism involves iron-related molecules [1]. It has been proven that other transition metals, such as cadmium, cobalt, copper, manganese, and zinc, are able to induce *Fpn* transcription [9], but the physiological relevance of such regulation remains largely unknown. Previous studies provided clear evidence of Fpn-mediated cellular efflux of iron, cobalt, and zinc, but not cadmium and copper [39]. Interestingly, it seems that significant differences exist regarding the mechanisms of transcriptional regulation of the *Fpn* gene by exposure to various transition metals. Metal transcription factor-1 (MTF-1) mediates the response to both metal excess and deprivation and is responsible for cellular protection against oxidative stress [40]. Zinc and cadmium induce *Fpn* transcription through the action of MTF-1, which is not the case for copper [9].

Interactions between copper and iron metabolism have been known for years and are mainly connected with processes of absorption, storage, and redistribution of those metals [11]. It is known that copper accumulates in the liver during iron deficiency and iron accumulates in the liver during copper deficiency [41,42]. To date, the molecular basis of those interplays still remains unknown. Here, we demonstrate that copper loading significantly increases the transcription of the *Fpn* gene, in both primary bone-marrow-derived macrophages (BMDMs) and the macrophage-like RAW 264.7 cell line. Importantly, this effect was not observed in BMDMs derived from *Nrf2* knockout mice, which indicates the role of this transcription factor in the process of Cu-dependent upregulation of *Fpn* transcription. Previously, the role of Nrf2 in heme-dependent ferroportin transcriptional induction via its binding to an antioxidant response element (ARE) sequence in the promoter of the *Fpn* gene was proven [5]. Herein, we are able to show that basal Fpn mRNA expression is also strictly dependent on Nrf2.

The ability of Cu to potentiate the activation of the Nrf2/ARE pathway in macrophages could be especially valuable in the context of the anti-inflammatory role of Nrf2, which contributes to the recruitment of inflammatory cells and regulation of gene expression [43]. Studies on copper-deficient animals clearly indicate the role of copper in immunomodulation [44]. Achard and co-workers showed that LPS and *Salmonella* infection increase the expression of genes responsible for copper uptake (*Ctr1)* and cellular copper redistribution (*Atp7a)* in macrophages [22]. Moreover, macrophage-mediated bacterial killing was shown to be strictly dependent on the presence of the Atp7a copper transporter on the phagosomal membrane [23], which could pump Cu into the lumen. Here, we also observed a significant induction of *Ctr1* and *Atp7a* expression in BMDMs, which confirms the ability of macrophages to alter their copper homeostasis under inflammatory conditions. On the other hand, iron handling within infected macrophages fits into the concept of “nutritional immunity”, where necessary elements such as iron are sequestered within cells (mainly macrophages) to reduce its availability to pathogens [45]. One of the mechanisms responsible for controlling iron accessibility during inflammation is the transcriptional repression of *Fpn* expression. This effect was observed in both in vivo and in vitro studies, where LPS treatment significantly decreased Fpn mRNA in macrophages [18,30,46,47].

When the iron sequestration process is prolonged, it can lead to hypoferremia and anemia of inflammation [16]; therefore, macrophages should release iron during the resolution phase of inflammation. Here, we observed that in both primary BMDMs and RAW 264.7 macrophages, LPS treatment significantly decreased Fpn mRNA levels, but concomitant CuCl_2_ stimulation was able to counteract the transcriptional suppression of *Fpn* and restore its expression to the level observed in the control group. It was previously reported that diethyl maleate and sulforaphane (both Nrf2 activators) are able to rescue *Fpn* expression after LPS treatment, but the mechanism remains unknown [19]. The increased expression of genes responsible for copper uptake and trafficking in macrophages after LPS treatment suggests that copper loading could enhance bacterial killing efficiency by generating reactive oxygen species (ROS) and influence the expression of Fpn, leading to scarce iron for intracellular pathogens via cytosolic iron depletion or increasing the iron pool inside the pathogen-containing cellular compartments to enhance ROS production. Recently, it has been proved that Fpn is required to provide iron to *Salmonella*-containing vacuoles (SCVs), where iron plays a key role as a co-factor in ROS generation [48]. A similar ROS-dependent mechanism driven by copper and the potential role of copper as an enhancer of Fpn expression under inflammatory states need to be confirmed and require further investigation.

Besides the transcriptional level, Fpn expression is controlled on the translational and systemic level. Here, in contrast to the transcriptional upregulation of *Fpn* after CuCl_2_ treatment, we did not observe significantly increased Fpn protein levels in the cellular extracts corresponding to the membrane fraction in primary macrophages. Interestingly, in our previous study focusing on the role of copper in maintaining iron homeostasis in *mosaic* mutant mice (*Atp7a* mutated gene leads to severe systemic disorder of copper metabolism), we observed a copper-dependent increase in both Fpn mRNA and protein abundance in bone-marrow-derived macrophages derived from wild-type control mice [49]. The discrepancy between studies regarding the regulation of Fpn after 16 h of CuCl_2_ stimulation is unclear; however, the use of different mouse strains (outbred colony vs. inbred C57Bl/6J) may contribute to the observed differences.

Moreover, here we observed that longer CuCl_2_ stimulation significantly decreased Fpn protein abundance, suggesting the involvement of cellular post-transcriptional or systemic post-translational control of Fpn expression. Hepcidin (*Hamp*) is the master regulator of iron homeostasis, which binds to Fpn and leads to its rapid internalization and degradation [4]. Hepcidin is mainly produced in the liver, but its expression increases in stimulated and pro-inflammatory macrophages and those infected with intracellular pathogens [35]. Our data confirm that primary bone-marrow-derived macrophages have the ability to produce hepcidin after LPS treatment, but copper exposure did not affect *Hamp* expression, suggesting that the latter mechanism is not responsible for the observed Fpn downregulation. The role of ceruloplasmin (Cp)—a copper-dependent ferroxidase that participates in the process of cellular iron efflux via the oxidation of Fe^2+^ to Fe^3+^—seems to be intriguing to reconcile as the potential factor contributing to the Fpn protein level. It has been shown previously that GPI-Cp (glycosylphosphatidylinositol (GPI)-anchored membrane form of Cp) is able to physically interact with Fpn localized within astrocyte cellular membranes, which in the absence of GPI-Cp was not able to efflux iron, and this suggests that Cp may impact the function or stability of Fpn [50]. However, these findings require further investigations.

Interestingly, copper loading did not alter macrophage ferritin levels (both light and heavy chains), suggesting that the decrease in Fpn was not sufficient to increase intracellular iron levels, at least to the extent required to stimulate ferritin synthesis. Moreover, CuCl_2_ treatment did not affect TfR1 mRNA expression, the level of which is highly dependent on cellular iron content and regulated by the interaction of IRPs with IRE sequences localized within the 3′ UTR of TfR1 mRNA [33]. These data are consistent with previous research on the J774 macrophage cell line showing that copper loading did not significantly influence TfR1 expression [10]. The IRE sequence is also localized within the 5 ’UTR mRNA of Fpn, which could impair *Fpn* expression through translational repression [3]. Our data suggest that macrophage copper loading does not modify the IRE binding activity of IRPs, which allows us to rule out this post-transcriptional regulatory mechanism as potentially being responsible for decreased Fpn protein levels.

Our analysis showed that the expression levels of selected genes involved in the processes of protein ubiquitination (*Usp14, Ubc*), proteasomes (*Psma1, Psmc4*), and the subset of endosomes that contain membrane-bound intraluminal vesicle formation (*Lip5,Chmp5*) were not significantly altered by copper loading. As lysosomal degradation is thought to be responsible for Fpn degradation [4], we treated BMDMs with chloroquine (Chlq) to impair lysosomal function. Chlq interferes with the activity of lysosomes, interacts with their membrane stability, and alters their signaling pathways [51]. Surprisingly, Chlq treatment led to a significant decrease in Fpn abundance in the membrane fraction of BMDMs. However, when combined with CuCl_2_, it increased the ferritin protein level in the cytosolic fraction, which suggests that the downregulatory effect is limited to Fpn. Interestingly, in hepatocytes exposed to elevated copper levels, one of the copper transporters, Atp7b, moves from the Golgi to lysosomes and imports metal into their lumen [52]. Moreover, in vivo models of copper overload demonstrated that increased copper loading in the liver leads to increased activity of lysosomal enzymes and reduces their structural integrity [53]. These data allow us to speculate that in macrophages, copper treatment and the disruption of lysosomal function may exert somewhat similar effects, affecting the stability of Fpn in an iron-independent manner.

Finally, it seems that cell type specificity may be responsible for the regulation of Fpn protein abundance in response to copper loading. In contrast to BMDMs, in primary peritoneal macrophages, Fpn levels did not change after copper treatment. Interestingly, in the RAW 264.7 macrophage-like cell line, copper exposure led to significant time-dependent increases in Fpn expression at both the mRNA and protein level. These data are consistent with the previously reported copper-dependent regulation of Fpn expression in macrophage-like cell line J774, where 48 h stimulation with CuSO_4_ was shown to significantly increase Fpn protein levels [10].

There are important differences between primary macrophages and macrophage cell lines regarding their responses to stimuli, which were previously reported in bone-marrow-derived macrophages and J774 cells infected with *Mycobacterium tuberculosis* [54]. Moreover, macrophages are considered as a heterogeneous cell population whose phenotype changes in response to their microenvironment and stimuli where they can adopt to a functional M1 (pro-inflammatory) or M2 (anti-inflammatory) program [55]. Bisgaard et al. (2016) had shown that the inflammatory response of BMDMs and peritoneal macrophages differs significantly, where the mRNA expression of M1/M2 markers was higher in peritoneal macrophages than BMDMs [56]. Moreover, it was shown previously that particular biometals such as copper in lower dosage (10 μM) are able to induce macrophage polarization towards an M2 phenotype, while higher concentrations (100 μM) stimulate a pro-inflammatory M1 response [57]. The M1/M2 macrophage phenotype impacts Fpn levels, where Fpn was downregulated in M1 and upregulated in M2 macrophages [58,59]. However, we did not observe a significant switch from M1 or M2 macrophage phenotypes after CuCl_2_ treatment (data not shown), suggesting that other factors are responsible for the observed differences.

In macrophages, Fpn is localized within lipid rafts, specific cell membrane compartments enriched in cholesterol and sphingolipids [25]. These membrane domains play a pivotal role in a variety of cellular processes, including intracellular trafficking and cell signaling [60,61,62]. It was previously shown that iron loading strongly enhances Fpn abundance in the low-density raft fraction of BMDMs and J774 macrophages, and raft integrity is important for the interaction of hepcidin with macrophage Fpn [25]. Here, we also report significant Fpn abundance in the lipid raft-enriched membrane fraction of RAW macrophages after iron loading, confirmed by the presence of a raft marker, flotilin 2 [27]. Interestingly, the subcellular pattern of Fpn localization in RAW macrophages after copper treatment differs from that obtained as a result of exposure to iron. Although some part of Fpn was still associated with lipid rafts, a significant amount was localized within dense, non-raft-enriched fractions (confirmed by the presence of a non-raft membrane protein, TfR1). Copper regulates genes involved in the cholesterol biosynthetic pathway [63] and is able to induce the redistribution of protein associated with the lipid raft fraction toward non-raft domains [64]. Therefore, because the hepcidin-mediated regulation of Fpn in macrophages strictly relies on lipid raft-dependent endocytosis, the possible copper-mediated influence on Fpn expression and cellular distribution seems to be interesting and worth investigating further.

## 5. Conclusions

In conclusion, we found that copper loading significantly enhances *Fpn* transcription in an Nrf2-dependent manner in primary bone-marrow-derived macrophages. Moreover, CuCl_2_ treatment affects Fpn expression in the RAW 264.7 macrophage-like cell line; however, cell-type-specific regulations seem to be involved in the processes connected to Fpn protein stability. In primary BMDMs, prolonged copper exposure led to a hepcidin- and iron-independent decrease in Fpn protein abundance, which was not the case for the macrophage cell line. The impact of copper loading on lysosomal activity in primary macrophages requires further investigation. In the context of the pivotal role of Fpn and iron trafficking within macrophages during infection, our data suggest that copper treatment is able to counteract the LPS-induced transcriptional suppression of Fpn expression, which may reflect additional immunomodulative properties of copper.

## Figures and Tables

**Figure 1 cells-10-02259-f001:**
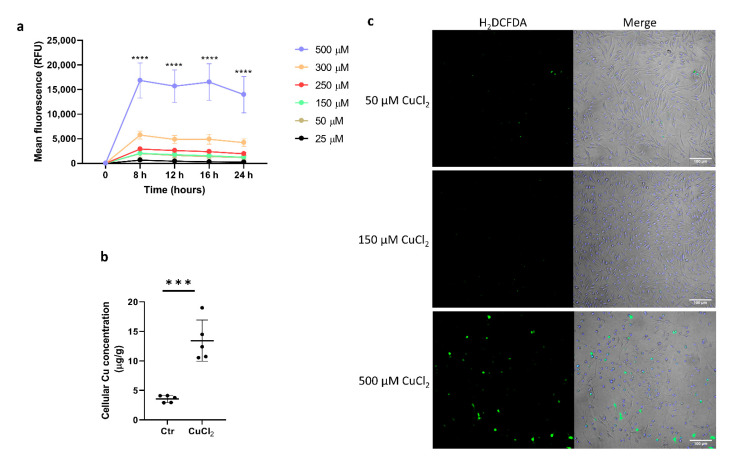
CuCl_2_ toxicity and ROS production in BMDMs. (**a**) Cytotoxic effect of CuCl_2_ treatment on BMDMs was measured in time (8–24 h) and dose (25–500 μM)-dependent manner. Fluorescence signal (RFU) produced by the dye binding to the dead-cell DNA is proportional to cytotoxicity. Ex: 485nm, Em: 520nm. Data were normalized to values obtained for non-treated control cells. Results are presented as mean ± SD, **** *p* < 0.0001. (**b**) Cellular copper concentration in BMDMs was determined by ICP-MS after 16 h incubation with 150 µM CuCl_2_. Data are presented as mean ± SD, *** *p* < 0.001. (**c**) ROS production: confocal images of BMDMs treated with 50, 150, and 500 µM CuCl_2_ loaded with 5 µM of ROS probe H_2_DCFDA. Hoechst was used as nucleus stain. Scale bar: 100 µm.

**Figure 2 cells-10-02259-f002:**
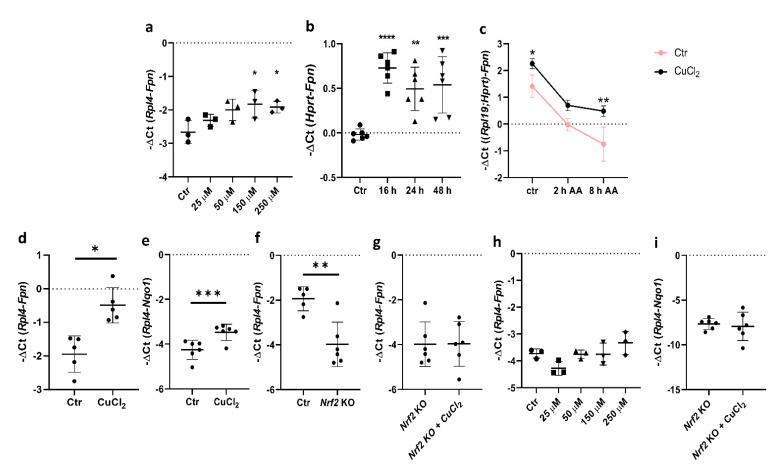
Copper loading affects *Fpn* transcription in BMDMs. (**a**) qPCR analysis of Fpn mRNA expression in BMDMs after 16 h exposure to various CuCl_2_ concentrations. Data represent mean ± SD,* *p* < 0.05. (**b**) qPCR analysis of Fpn mRNA expression in BMDMs after exposure to 150 μM CuCl_2_ for 16, 24, and 48 h. Data represent mean ± SD, ** *p* < 0.01, *** *p* < 0.001, **** *p* < 0.0001. (**c**) qPCR analysis of Fpn mRNA expression in BMDMs after 16 h exposure to 150 μM CuCl_2_ followed by 2 and 8 h incubation with 2 μg/mL α-amanitin. Data were normalized to expression level of two reference genes, *Rpl19* and *Hprt.* Results presented as mean ± SD, * *p* < 0.05, ** *p* < 0.01. (**d**,**e**) qPCR analysis of Fpn and Nqo1 mRNA expression in BMDMs after 16 h exposure to 150 μM CuCl_2_. Data represent mean ± SD,* *p* < 0.05,*** *p* < 0.001. (**f**) qPCR analysis of *Fpn* mRNA expression in BMDMs derived from wild-type and *Nrf2* knockout mice after 16 h exposure to 150 μM CuCl_2_. Data represent mean ± SD, ** *p* < 0.01. (**g–i**) qPCR analysis of Fpn and Nqo1 mRNA expression in BMDMs derived from *Nrf2* knockout mice after 16 h exposure to 150 μM CuCl_2_. Data represent mean ± SD. Graph in (**h**) reflects Fpn mRNA expression level in *Nrf2* knockout BMDMs after 16 h exposure to various CuCl_2_ concentrations.

**Figure 3 cells-10-02259-f003:**
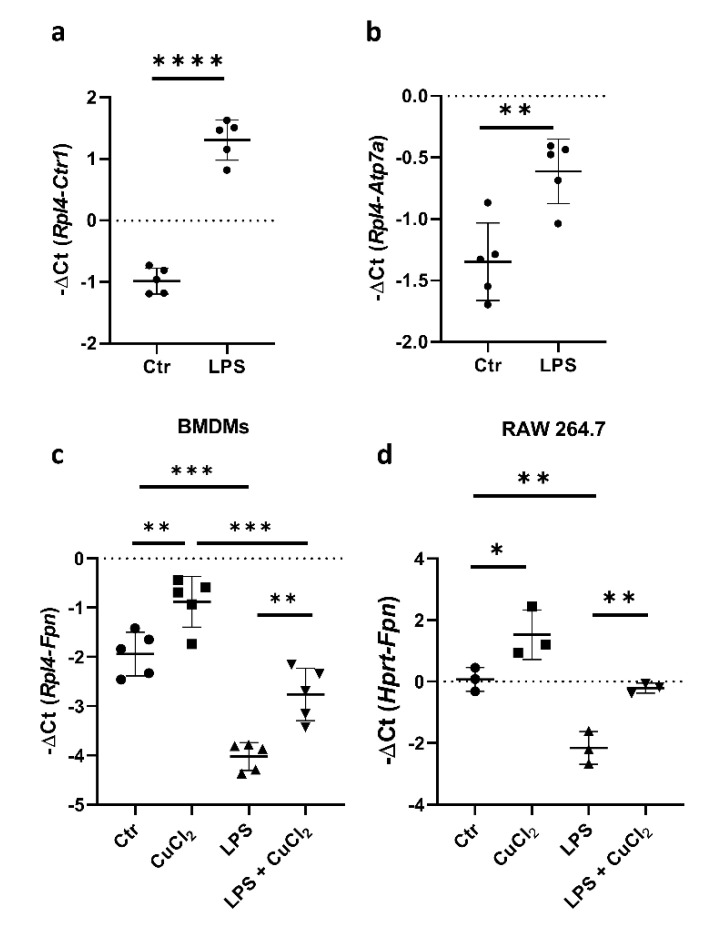
Copper loading counterweights LPS-induced suppression of Fpn mRNA. (**a**,**b**) qPCR analysis of Ctr1 and Atp7a mRNA expression in BMDMs incubated with 100 ng/mL lipopolysaccharide (LPS) for 16 h. Data represent mean ± SD, ** *p* < 0.01, **** *p* < 0.0001. qPCR analysis of Fpn mRNA expression in (**c**) BMDMs and (**d**) RAW 264.7 macrophages treated with 100 or 200 ng/mL LPS, respectively, in the absence or presence of 150 μM CuCl_2_ for 16 h. Data represent mean ± SD, * *p* < 0.05, ** *p* < 0.01, *** *p* < 0.001.

**Figure 4 cells-10-02259-f004:**
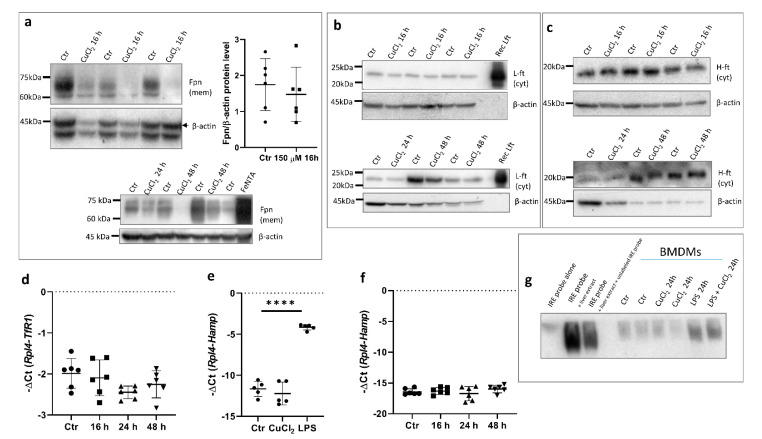
Copper loading decreases Fpn protein levels in hepcidin- and IRE-independent manner. (**a**) Upper panel: Western blot analysis of Fpn protein level in membrane fraction of control cells and BMDMs treated with 150 μM CuCl_2_ for 16 h. Graph illustrates densitometric analysis of Fpn protein level normalized to β-actin level from 6 independent experiments. Bottom panel: Western blot analysis of Fpn protein level in membrane fraction of control cells and BMDMs treated with 150 μM CuCl_2_ for 24 and 48 h. BMDMs stimulated with 150 μM FeNTA were used as positive controls to check specificity of antibody. (**b**) Western blot analysis of L-ferritin protein level in cytosolic fraction of control cells and 150 μM CuCl_2_-treated BMDMs. Recombinant mouse ferritin light chain (Rec L-Ft) was used as positive control. (**c**) Western blot analysis of H-ferritin protein levels in cytosolic fraction of control and 150 μM CuCl_2_-treated BMDMs. (**d**) qPCR analysis of TfR1 mRNA expression in BMDMs incubated with 150 μM CuCl_2_ for 16, 24, and 48 h. Data represent mean ± SD. (**e**) qPCR analysis of *Hamp* expression in BMDMs incubated with 150 μM CuCl_2_ or 100 ng/mL lipopolysaccharide (LPS) for 16 h. Data represent mean ± SD, **** *p* < 0.0001. (**f**) qPCR analysis of Hamp mRNA expression in BMDMs incubated with 150 μM CuCl_2_ for 16, 24, and 48 h. Data represent mean ± SD. (**g**) RNA-EMSA was performed to assess IRE-IRP binding activity in BMDMs treated with 150 µM CuCl_2_ or 200 ng/mL LPS for 24 h. Cytosolic extract (20 µg) from BDMDs was incubated with biotinylated IRE probe, and RNA-protein complexes were separated on non-denaturing polyacrylamide gel. Biotin-IRE control RNA, biotin-IRE control RNA + cytosolic extract from liver (5 µg), and biotin-IRE control RNA + cytosolic extract from liver (5 µg) + 200-fold molar excess of unlabeled IRE RNA were used as positive control reaction.

**Figure 5 cells-10-02259-f005:**
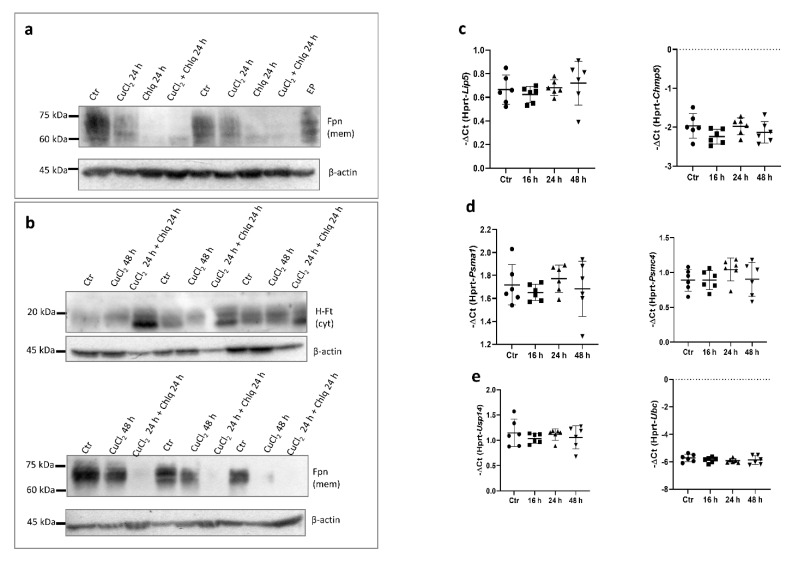
Impact of copper loading on mechanisms controlling Fpn protein degradation in BMDMs. (**a**) Western blot analysis of Fpn protein level in membrane fraction of BMDMs treated with 150 μM CuCl_2_, 100 μM chloroquine, or combination of both for 24 h. Extract derived from BMDMs after 6 h of erythrophagocytosis (EP) was used as a positive control. (**b**) Western blot analysis of H-ferritin protein level in cytosolic fraction (upper panel) and Fpn protein level in membrane fraction (bottom panel) of BMDMs treated with 150 μM CuCl_2_ for 48 h or pretreated with 150 μM CuCl_2_ for 24 h followed by 100 μM chloroquine stimulation for remaining 24 h. (**c**–**e**) qPCR analysis of Lip5, Chmp5, Psma1, Psmc4, Usp14, and Ubc mRNA expression in BMDMs incubated with 150 μM CuCl_2_ for 16, 24, and 48 h. Data represent mean ± SD.

**Figure 6 cells-10-02259-f006:**
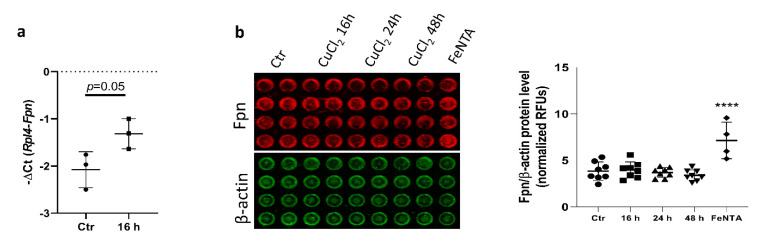
CuCl_2_ treatment does not affect Fpn mRNA and protein levels in primary peritoneal macrophages. (**a**) qPCR analysis of Fpn mRNA expression in primary peritoneal macrophages incubated with 150 μM CuCl_2_ for 16 h. Data represent mean ± SD. (**b**) In-Cell Western blot detection (left panel) and quantification (histogram) of Fpn expression in peritoneal macrophages treated with 150 µM CuCl_2_ for 16, 24, and 48 h. Cells treated with Fe-NTA (200 µM) were used as positive control. Fpn intensity signal was normalized to β-actin level. Data on histogram represent mean ± SD, **** *p* < 0.0001.

**Figure 7 cells-10-02259-f007:**
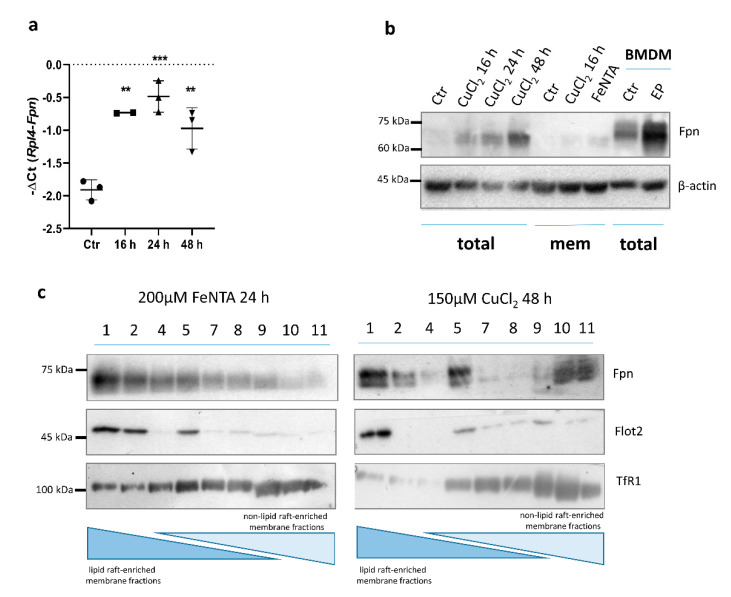
Copper loading increases Fpn mRNA level. (**a**) qPCR analysis of Fpn expression in RAW 264.7 cells incubated with 150 μM CuCl_2_ for 16, 24, and 48 h. Data represent mean ± SD, ** *p* < 0.01, *** *p* < 0.001. (**b**) Western blot analysis of Fpn protein level in total extract and membrane fraction of control and 150 μM CuCl_2_-treated RAW 264.7 cells. Extract derived from BMDMs after 6 h of erythrophagocytosis (EP) was used as positive. (**c**) Western blot detection of Fpn and flotilin 2 (Flot2; raft marker) in lipid raft and non-raft-enriched (TfR1) membrane fractions of Fe-NTA (200 μM) and CuCl_2_ (150 μM) treated RAW 264.7 macrophages. Numbers indicate subsequent fractions collected after discontinuous iodixanol density gradient ultracentrifugation.

## Data Availability

All data supporting the findings of this research are available within the article or from the corresponding author upon request.

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
