# Peer review of "The Role of Copper in the Regulation of Ferroportin Expression in Macrophages"

_cells, 2021, doi:10.3390/cells10092259_

Round 1

Reviewer 1 Report

In this paper the authors investigate the effect of copper on expression of the iron exporter ferroportin, both at the mRNA and protein level. While an effect of copper on Fpn was described over ten years ago (as correctly reported by the authors), the authors present many new interesting data on the molecular mechanisms underlying regulation of expression of Fpn by copper.  A few points should be addressed to better clarify their findings: the finding that expression of Fpn protein decreases in BMDMs after 16 h treatment with 150 uM CuCl2 but it increases in RAW264.7 cells and it is unchanged in primary peritoneal macrophages could be discussed also in terms of the M1/M2 phenotype of the different cells, a factor which impacts Fpn levels. The connection of copper-dependent ferroxidases for Fpn stability should be mentioned, at least by adding a sentence in the discussion.

Minor points:

1) in figure 7c the lipid raft separation is not particularly convincing with TfR1 distributed throughout the gradient, it may be better described as an enrichment rather than a fractionation

2) improve legend of figure 1 by adding how cell viability was measured (panel a), this is only indicated in the methods section

3) check references: many are incomplete

4) there are a few typos

Author Response

Dear Reviewer,

I uploaded my responses to your comments as a Word/PDF file.

Reviewer 2 Report

The manuscript of Aneta Jończy et al., demonstrates that copper loading significantly enhances Fpn transcription in macrophages while Fpn protein abundance in response to CuCl2 treatment depends on macrophage type and factors specific to macrophage population that under this situation can influence the Fpn regulation in response to copper loading. The role of cooper is intriguing as in a number of diseases (Wilson’s, Menkes, or viral infections) homeostasis of cooper is dysregulated.

In response to cooper treatment, what happens to MTF-1 transcription factor binding?? It is known that upon zinc treatment MTF-1 transcription factor binds to three consensus MRE cis-acting elements in the 1.8 kb HAMP gene promoter, with the first one consisting of two overlapping elements in reverse orientation, thereby modulating hepcidin promoter activity. The work is very well presented and results are clearly processed.

Minor points:

  • It is well known that Hepcidin is the only recognized ligand for FPN and regulates FPN activity by inducing its internalization and proteolysis (Nemeth et al., 2004). Could HAMP gene expression by acting as a zinc ion ‘sponge’, thereby causing imbalance in the MTF-1/MRE/HAMP regulatory axis result in enhancement of Fpn transcription in macrophages?? Please see for this phenomenon (Dimitriadis et al., 2021).

  • Fig. 2 : In Copper loading affects Fpn transcription in BMDMs. (a) qPCR analysis of Fpn mRNA expression in BMDMs after 16h exposure to varied CuCl2 concentrations. Why 16h ? what happens in early hours ? The authors should present data for 6 or 8h. What happens in RAW264.7 cells?

  • What is the cooper buffering capacity of BMDMs? Does this cell line constitutively express metallothionein‐1 (MT‐1), a protein that plays a key role in the protection against metal toxicity and oxidative stress, through its ability to bind both physiological and xenobiotic heavy metals and ions?

Author Response

Dear Reviewer,

I uploaded my responses to your comments as a Word/PDF file

Reviewer 3 Report

This is an interesting manuscript which really needs serious attention to the English, for  example (changes in red):

466   When the iron sequestration process is prolonged, it can lead to  hypoferremia and anemia of inflammation

479    its expression increases  in stimulated, pro-inflammatory macrophages  and in macrophages -infected with intracellular pathogens 

485    that the decrease in Fpn was not sufficient enough to increase intracellular iron levels, at least to the extent able required to stimulate ferritin synthesis

There are many, many more throughout the entire text, and their correction would enable the article to be aceptable for publication

Author Response

Dear Reviewer,

Thank you for your revision of our manuscript. Our manuscript was sent to MDPI English editing  platform.